# Site-Specific Radiolabeling of a Human PD-L1 Nanobody via Maleimide–Cysteine Chemistry

**DOI:** 10.3390/ph14060550

**Published:** 2021-06-08

**Authors:** Dora Mugoli Chigoho, Quentin Lecocq, Robin Maximilian Awad, Karine Breckpot, Nick Devoogdt, Marleen Keyaerts, Vicky Caveliers, Catarina Xavier, Jessica Bridoux

**Affiliations:** 1In Vivo Cellular and Molecular Imaging Laboratory (ICMI), Medical Imaging Department (MIMA), Vrije Universiteit Brussel, 1090 Brussels, Belgium; dora.mugoli.chigoho@vub.be (D.M.C.); nick.devoogdt@vub.be (N.D.); marleen.keyaerts@vub.be (M.K.); vicky.caveliers@vub.be (V.C.); catarina.xavier@vub.be (C.X.); 2Laboratory for Molecular and Cellular Therapy (LMCT), Department of Biomedical Sciences, Vrije Universiteit Brussel, 1090 Brussels, Belgium; quentin.lecocq@vub.be (Q.L.); robin.maximilian.awad@vub.be (R.M.A.); karine.breckpot@vub.be (K.B.); 3Department of Nuclear Medicine, UZ Brussel, Laarbeeklaan 101, 1090 Brussels, Belgium

**Keywords:** cancer, nanobodies, PD-L1, site-specific, PET, gallium-68, maleimide-NOTA

## Abstract

Immune checkpoint inhibitors targeting the programmed cell death-1 (PD-1) and its ligand PD-L1 have proven to be efficient cancer therapies in a subset of patients. From all the patients with various cancer types, only 20% have a positive response. Being able to distinguish patients that do express PD-1/PD-L1 from patients that do not allows patients to benefit from a more personalized and efficient treatment of tumor lesion(s). Expression of PD-1 and PD-L1 is typically assessed via immunohistochemical detection in a tumor biopsy. However, this method does not take in account the expression heterogeneity within the lesion, nor the possible metastasis. To visualize whole-body PD-L1 expression by PET imaging, we developed a nanobody-based radio-immunotracer targeting PD-L1 site-specifically labeled with gallium-68. The cysteine-tagged nanobody was site-specifically conjugated with a maleimide (mal)-NOTA chelator and radiolabeling was tested at different nanobody concentrations and temperatures. Affinity and specificity of the tracer, referred to as [^68^Ga]Ga-NOTA-mal-hPD-L1 Nb, were assayed by surface plasmon resonance and on PD-L1^POS^ or PD-L1^NEG^ 624-MEL cells. Xenografted athymic nude mice bearing 624-MEL PD-L1^POS^ or PD-L1^NEG^ tumors were injected with the tracer and ex vivo biodistribution was performed 1 h 20 min post-injection. Ideal ^68^Ga-labeling conditions were found at 50 °C for 15 min. [^68^Ga]Ga-NOTA-mal-hPD-L1 Nb was obtained in 80 ± 5% DC-RCY with a RCP > 99%, and was stable in injection buffer and human serum up to 3 h (>99% RCP). The in vitro characterization showed that the NOTA-functionalized Nb retained its affinity and specificity. Ex vivo biodistribution revealed a tracer uptake of 1.86 ± 0.67% IA/g in the positive tumors compared with 0.42 ± 0.04% IA/g in the negative tumors. Low background uptake was measured in the other organs and tissues, except for the kidneys and bladder, due to the expected excretion route of Nbs. The data obtained show that the site-specific ^68^Ga-labeled NOTA-mal-hPD-L1 Nb is a promising PET radio-immunotracer due to its ease of production, stability and specificity for PD-L1.

## 1. Introduction

Immune responses are managed through activatory and inhibitory signals at different checkpoints. Programmed cell death-protein-1 (PD-1) and its ligands, programmed death receptor ligands 1 and 2 (PD-L1, PD-L2), are a part of the inhibitory checkpoints modulating the activity of T lymphocytes [1,2]. Cytotoxic T cells have the ability to recognize and eliminate cancer cells. PD-L1 expression on healthy cells is a mechanism to prevent autoimmunity [3,4], and some cancer cells evade the anti-tumor immune response by also expressing PD-L1. Monoclonal antibody-based immune checkpoints inhibitors (ICIs) such as Pembrolizumab, Nivolumab, Atezolizumab and Durvalumab have been FDA approved to block the PD-1:PD-L1 interactions and have shown successful results in cancer patients [5,6]. Despite encouraging results, approximately four patients out of five are not responsive to the PD-1:PD-L1 inhibitors alone, limiting the clinical potential of the treatment [7,8]. Screening the patients for PD-L1 expression levels in their lesions before starting the treatment is one way to tackle this obstacle. To predict the ICI treatment response, a biopsy of one of the tumor lesions is examined through immunohistochemical detection for PD-1 or PD-L1. However, this technique does not represent the heterogeneity of PD-L1 expression within the entire patient’s body [9]. Positron emission tomography (PET) imaging could overcome the current limitations by providing molecular information of the whole body in a non-invasive manner. In nuclear medicine, PET is the most preferred imaging technique due to its sensitivity and quantitative accuracy [10,11].

In previous studies, we selected a lead high affinity PD-L1 nanobody (Nb) which was labeled with gallium-68 (^68^Ga, T_1/2_ = 68 min) that allowed non-invasive detection of PD-L1 positive tumor in xenografted mice [12,13]. The Nb was randomly conjugated with the *p*-NCS-Bn-NOTA bifunctional chelating agent on its lysines. As a random strategy may impact the biodistribution of a tracer, the randomly radiolabeled Nb was compared to a site-specifically radiolabeled analogue. This approach involved the sortase-A enzyme that allowed us to couple the NOTA chelator at the *C*-terminus of the Nb. The two techniques resulted in similar yields, purity and similar in vivo behaviors, showing that the random strategy is a valid method for clinical translation of the ^68^Ga-labeled NOTA-(hPD-L1) Nb. However, site-specific strategies offer the advantage of being more reproducible and yielding a homogenous pharmaceutical product that can be well characterized, which is not the case for random conjugations [14,15]. For these reasons, site-specific strategies are preferred for clinical translation. In the case of the sortase-mediated reaction, however, the enzymatic process would represent a challenge for clinical translation in terms of qualification and validation of the starting materials, production process and identification of process-related impurities. Examples are the quality requirements of the enzyme, the custom production of the acceptor peptide containing NOTA and the additional purification steps such as immobilized metal affinity chromatography (IMAC) using Nickel–Nitrilotriacetic acid (Ni-NTA) beads. These additions require extra quality controls, such as quantification of residual enzyme or nickel in the final product.

In this study, we investigate an alternative and more straightforward site-specific strategy to produce a ^68^Ga-labeled hPD-L1 Nb via maleimide–cysteine (mal–Cys) chemistry (Figure 1). Mal–Cys bioconjugations are already well implemented and mal-modified chelators such as mal-NOTA are commercially available [16]. The Nb can be engineered with a *C*-terminal Cys-tag, typically resulting in the formation of Nb-dimers and monomers capped with glutathione. Both can be reduced under mild conditions in order to site-specifically couple mal-NOTA, leaving intact the internal cysteines that are necessary to maintain the Nb’s tertiary structure [17].

## 2. Results

### 2.1. Nanobody Functionalization, Characterization and Affinity Analysis

From the periplasmic extracts of transformed and shake-flask cultured *E. coli* bacteria grown, we recovered 20 mg of Cys-tagged Nb product, consisting of a mixture of dimer and monomer (32328 Da, 15590 Da, respectively; final yield = 5.8 mg/L culture). After 90 min of incubation with the mild reducing agent 2-mercaptoethylamine (2-MEA), more than 95% of the Nb was converted to monomeric Nb, as observed by size-exclusion chromatography (SEC).

The reduction step was followed by site-specific coupling of mal-NOTA on the free thiol of the *C*-terminal cysteine. SEC purification of the conjugated Nb resulted in an average recovery yield of 62 ± 6% (*n* = 4) from the conjugation procedure. Quality control (QC) by sodium dodecyl sulfate polyacrylamide gel electrophoresis (SDS-PAGE) and SEC revealed a purity >95%, as depicted in Appendix A. Electrospray-ionization quadrupole time-of-flight mass spectrometry (ESI-Q-ToF) of the site-specifically modified NOTA-mal-Nb showed a major peak of NOTA-mal-Nb (15588 Da) and deamidated NOTA-mal-Nb (15571 Da), as shown in Appendix A. Starting dimeric Nb or monomeric Nb were not observed. The melting point of NOTA-mal-Nb was determined via a protein melting program of a RealTime PCR instrumentation and measured at 75 °C.

The affinity kinetics with conjugated and unconjugated Nbs were measured by surface plasmon resonance (SPR) on immobilized hPD-L1 recombinant protein. Modified and unmodified Nbs exhibited an equilibrium dissociation constant (K_D_) in the same range, namely 4.38 nM (*n* = 2) and 2.1 nM, respectively, suggesting that the procedure did not impact the affinity of the Nbs.

### 2.2. Radiolabeling and In Vitro Stability Studies

In order to optimize the labeling of NOTA-mal-hPD-L1 Nbs with ^68^Ga, different parameters, such as the Nb concentration, temperature and incubation time, were evaluated. Increasing the temperature while keeping a constant Nb concentration of 3.6 μM had a remarkable impact on radiochemical purity (RCP), as depicted in Figure 2A. Likewise, the concentration of the Nb in the reaction mixture, ranging from 2.9 to 4.2 µM, was tested at 50 °C. Increasing the concentration above 3.6 μM did not lead to an improved RCP, as depicted in Figure 2B. After optimization, an average RCP > 90% (before purification) could be obtained using a Nb concentration of 3.6 µM and 15 min incubation at 50 °C. After purification, RCP was higher than 99%, with an 80 ± 5% radiochemical purity decay-corrected DC-RCY (*n* = 2).

The stability of the radiolabeled Nb was tested in injection buffer and in human serum at 37 °C. A RCP of > 95% was retained up to 180 min post-labeling in injection buffer and >85% in human serum, as shown by radio-SEC in Appendix A.

In addition, the stability of the NOTA-mal-(hPD-L1) Nb precursor was also followed. After two months of storage at −20 °C in 0.1 M NH_4_OAc pH 7, SEC analysis of the NOTA-mal-hPD-L1 Nb showed >98% purity (Appendix A). Radiolabeling and stability results also remained constant. After 15 min incubation at 50 °C, an RCP of >95% was measured by radio-instant thin layer chromatography (radio-iTLC) and RCP remained at >99% after purification in injection buffer over 180 min (Appendix A).

### 2.3. Cell Binding Study

To assess the specificity of the site-specific NOTA-coupled Nb to bind to hPD-L1 expressed on cells, it was labeled with ^68^Ga and added to either hPD-L1^POS^ or hPD-L1^NEG^ 624-MEL cells in either 3 or 15 nM concentrations. After incubation, the unbound fraction was washed away, and the cell-associated activity was measured. The percentage of cell-associated activity of the radiolabeled Nb showed specific binding on hPD-L1^POS^ cells, which was confirmed by absence of cell-associated activity in control conditions (hPD-L1^NEG^ cells and excess of unlabeled Nb). For the 3 nM concentration, a significantly higher amount of bound activity on hPD-L1^POS^ cells than on hPD-L1^NEG^ cells was measured (*p* = 0.0009), as well as on blocked cells (*p* = 0.0009), as represented in Figure 3 (3.94 ± 0.73% vs. 0.25 ± 0.03% vs. 0.21 ± 0.06%, respectively, *n* = 3). When increasing the concentration of Nb from 3 nM to 15 nM on cells, the percentage of cell-associated activity was lower (3.94 ± 0.73% vs. 1.07 ± 0.35%, respectively, *p* = 0.0154), which could demonstrate that the fraction of unlabeled Nb starts competing with the fraction of ^68^Ga-labeled Nb. As a result, differences between the percentage of cell-associated activity at 15 nM on hPD-L1^POS^ and hPD-L1^NEG^ cells, as well as in blocking conditions (1.07 ± 0.35% vs. 0.19 ± 0.01% vs. 0.19 ± 0.05%, respectively, *n* = 2), were not significant (*p* = 0.0709 and *p* = 0.0720, respectively).

### 2.4. Biodistribution and In Vivo Tumor Targeting

The final step consisted of investigating whether the [^68^Ga]Ga-NOTA-mal-hPD-L1 Nb tracer was able to target hPD-L1 positive tumors in vivo. Organ and tissue biodistribution results at 80 min post-injection (p.i.) are represented in Figure 4 and Appendix A. The ^68^Ga-labeled Nb showed higher uptake of 1.86 ± 0.67% IA/g (*n* = 12) in the positive tumors compared with 0.42 ± 0.03% IA/g (*n* = 6) in the negative tumors, which is significantly different (*p* = 0.0002). Kidney uptake was 27.9 ± 5.1% IA/g (*n* = 12) in the group bearing hPD-L1^POS^ tumors. All other organs were at background level, as depicted in Figure 4. An average tumor-to-blood ratio of 5.17 ± 1.82% (*n* = 12) and an average tumor-to-muscle ratio of 27.70 ± 12.59% (*n* = 12) were measured for the group bearing hPD-L1^POS^ tumors. Flow cytometry (FC) analysis confirmed the hPD-L1 expression of the positive cells in tumors compared with the negative tumors (Appendix A).

## 3. Discussion

Immune checkpoint inhibitors blocking the mechanisms exploited by cancer cells to evade the immune system have proven to be a successful approach in treating cancer. Differences in response rate to PD-L1 inhibitor treatments are observed amongst patients, making the development of predictive markers helpful for identifying patients who are most likely to benefit from such treatments. Developing a PET-tracer would allow us to estimate the PD-L1 expression in the tumor lesions in a non-invasive and reproducible way.

We previously developed a hPD-L1 Nb that was functionalized with the NOTA chelator via conjugation to the accessible lysines, and efficiently radiolabeled with ^68^Ga for PET imaging. The Nb was also site-specifically functionalized and radiolabeled using the sortase-A enzymatic approach [12,13,18]. In the current study, we aimed to site-specifically radiolabel the hPD-L1 Nb via an alternative chemical strategy that does not require the use of an enzyme. Cysteine–maleimide couplings are a popular and straightforward alternative for functionalizing proteins and have already made their way to the clinic. One promising example is the antibody–drug conjugate (ADC) against *S. aureus* used in phase I clinical trials, for which the drug was conjugated to the monoclonal antibody via a maleimide–thiol coupling, yielding a homogeneous ADC product with improved therapeutic potential [19,20]. Compared to click-chemistry, which is a more recent technique for site-specific coupling between proteins and moieties, mal–Cys couplings show some advantages, such as the ease of introducing a free cysteine in the protein structure compared with the introduction of a click-reactive group.

In particular, this technique can be applied to functionalize Nbs bearing a free cysteine in a His_6_-linker-Cys-tag at their *C*-terminus [14,16]. One known main disadvantage in the production of Cys-tagged Nbs is an average of 50% loss in production yield compared to productions of His_6_-tag only containing Nbs [14]. In the case of the hPD-L1 Nb, the production yield remained comparable to the sortag-His_6_-tag Nb’s production and was about 45% lower than the His_6_-tag Nb’s production (5.8 mg/L (*n* = 1) vs. 4.5 ± 0.9 mg/L (*n* = 2) vs. 11.2 ± 9.9 mg/L (*n* = 2), respectively). Maleimide-NOTA was site-specifically coupled to the hPD-L1 Nb bearing the Cys-tag, in similar recovery yields as for a random coupling on the lysines. Although, a cysteine–maleimide coupling requires an extra mild reduction step to ensure the thiol function of cysteine is free and no dimer and monomer capped with a glutathione are present. The reaction time remains similar as for the random coupling and is lower for the sortase-A mediated coupling (3 h 30 min vs. 2 h 30 min vs. 16 h, respectively).

^68^Ga-labeling of NOTA-mal-(hPD-L1) Nb required heating at 50 °C to reach similar DC-RCY as for the NCS-NOTA-coupled Nb (80% at 50 °C for 15 min vs. 86% at RT for 10 min, respectively), due to the difference in structure between the mal-NOTA and the NCS-NOTA used for the random coupling [13]. The latter possesses an isothiocyanate (NCS) function attached to its backbone structure and is not interfering with the chelation capacity of the three free carboxylic arms, while the maleimide function on mal-NOTA is attached to one of the arms bearing the carboxylic function. The functionalization on the arm is most likely the reason that temperature is required to increase the radiolabeling reaction kinetics, as reported already with ^111^In and ^64^Cu [21,22]. This elevated temperature is not an issue for the stability of NOTA-mal-Nb, for which a melting temperature of 75 °C was measured. In addition, the mal-NOTA-functionalized Nb did not show affinity loss, as measured by SPR, and mass spectrometry (MS) analysis confirmed that no free cysteine-containing starting Nb remained after purification.

[^68^Ga]Ga-NOTA-mal-(hPD-L1) Nbs proved to be stable in injection buffer and in human serum up to 3 h of incubation. In vitro cell binding studies confirmed the functionality and specificity of [^68^Ga]Ga-NOTA-mal-(hPD-L1) Nbs. Stability during storage of the NOTA-mal-(hPD-L1) Nb at −20 °C in 0.1 M NH_4_OAc pH 7 was tested over two months. ^68^Ga-labeling remained equally efficient over time. Stability of the labeled compound in injection buffer, produced with the two months old NOTA-mal-(hPD-L1) Nb, showed that this radiolabeled Nb could also remain stable. These results are promising for clinical practice since the NOTA-conjugated Nb may be stored for long periods before being radiolabeled and used, although later time points remain to be tested.

In vivo biodistribution showed that kidney retention was higher than for the randomly coupled, ^68^Ga-labeled Nb analogue, as well as for the sortase-A mediated site-specifically NOTA-coupled hPD-L1 Nb (27.9 ± 5.1% IA/g vs. 13.8 ± 2.7% IA/g vs. 8.2 ± 1.9% IA/g). The Cys-tagged NOTA-mal-(hPD-L1) Nb contains a hexahistidine and a rigid linker of 14 amino acids (AAs), necessary to minimize disturbance during the bacterial expression to produce the Nb [14,23]. This extra tag results in an increase in overall charges, leading to higher kidney retention compared with the random NOTA-Nb, which only contains a His_6_-tag or the sortase-A mediated NOTA-coupled Nb, for which the sortag-His_6_-tag is cleaved during coupling [13,24].

The cysteine–maleimide linkage is known to be unstable in vivo due to the competition of thiol-containing proteins and irreversible hydrolysis [25]. These effects are often observed a day to several weeks after injection [26]. Therefore, this issue is not of concern when using a ^68^Ga-labeled NOTA-mal-Nb, since imaging is possible as early as 1 h p.i. In addition, the biodistribution profile at 80 min p.i. did not show any uptake in organs or expected blood retention, confirming that no radio-metabolite reactive with thiol-containing protein was formed during this timeframe.

The tumor uptake of [^68^Ga]Ga-NOTA-mal-(hPD-L1) Nbs was significantly higher in the mice with hPD-L1^POS^ tumors than with hPD-L1^NEG^ tumors (1.86 ± 0.67% IA/g vs. 0.42 ± 0.04% IA/g, respectively). The hPD-L1 expression in the dissected tumors was confirmed by FC. In the same tumor model, the tumor uptake was as similar to the currently reported [^68^Ga]Ga-NOTA-mal-(hPD-L1) as it was for the previously reported randomly and sortase-A mediated NOTA-coupled ^68^Ga-labeled Nb analogues (1.86 ± 0.67% IA/g, 1.89 ± 0.40% IA/g and 1.77 ± 0.28% IA/g, respectively, non-significant (NS)) [13]. Tumor-to-blood (T/B) and tumor-to-muscle (T/M) ratios were also similar to the two previously reported analogues (T/B: 5.17 ± 1.82% vs. 5.37 ± 1.49% vs. 6.28 ± 2.95%, respectively, NS; T/M: 27.70 ± 12.59% vs. 28.00 ± 10.62% vs. 34.53 ± 13.24%, respectively, NS) [13]. These results support that this site-specifically ^68^Ga-labeled hPD-L1 Nb analogue is as efficient as the randomly and sortase-A mediated NOTA-Nb analogues. The cysteine–maleimide coupling reaction is as straightforward as the random coupling, with the advantage of yielding a homogenous site-specifically coupled NOTA-Nb. In vivo, [^68^Ga]Ga-NOTA-mal-(hPD-L1) Nb demonstrated a similar behavior to the two other analogues. One known main disadvantage is the loss in production efficiency compared with productions of His_6_-tag-containing only Nbs used in random labeling, although efficiency is comparable with the sortag-His_6_-tag-bearing Nb. Before clinical translation, the full His_6_-linker-Cys-tag should be optimized. This should be performed to, on one hand, reduce kidney retention (by removing the His_6_-tag or modifying the AAs in the linker) and, on the other hand, to improve the production yield by optimizing the linker (length and AAs). Finally, the process may be optimized for GMP production in yeast or *E. coli*.

## 4. Materials and Methods

### 4.1. Purification of hPD-L1 from Perisplasmic Extract

The periplasmic extracts (PE) containing the (hPD-L1)-cysteine-tagged nanobody (Cys-tagged-hPD-L1 Nb) were produced and provided in collaboration with Cellular and Molecular Immunology (CMIM), *Vrije Universiteit Brussel*, Belgium. IMAC 250 μL Nickel beads’ NTA-resin (Thermo Fisher Scientific, Merelbeke, Belgium) per 50 mL of periplasmic extract was added, and the mixture was shaken for 1 h. The mixture was centrifuged at 1400 rpm for 5 min and the supernatant was replaced by PBS. The periplasmic extract (PE) was shaken again for 1 h, centrifuged and the supernatant was discarded.

The reaction mixture was resuspended in PBS (10 mL) and filtrated on a PD-10 column (GE healthcare, 14.5 mm × 50 mm, 8.3 mL) using freshly made imidazole buffer (0.5 M, pH 7–7.4, filtered through a 0.22 μm filter) to eluate.

### 4.2. Chromatographic Analysis

SEC columns were purchased from GE Healthcare (Diegem, Belgium). SEC purification of Nb isolated from the PE was performed on a HiLoad 16/600 Superdex 30 pg column using metal free phosphate buffer saline (1× PBS: 2.68 mM KCl, 137 mM NaCl, 1.47 mM KH_2_PO_4_, 8.1 mM Na_2_HPO_4_) at a flow rate of 1 mL/min. The SEC purification and QC analyses of the site-specifically functionalized Nb were performed on a Superdex 75 Increase 10/300 GL column using 0.1 M NH_4_OAc pH 7, at a flow rate of 0.8 mL/min. For QC, RCP was also assayed with binderless glass microfiber paper that was impregnated with silica gel (iTLC-SG) (Agilent Technologies, Diegem, Belgium) using 0.1 M sodium citrate buffer pH 4.5–5 as eluent. Serum samples were analyzed by SEC on a Superdex 5/150 GL using 2× PBS at a flow rate of 0.3 mL/min.

### 4.3. Production and Purification of the Cysteine-Tagged hPD-L1 Nb

The (hPD-L1)-Cys-tag Nb was produced in *E. coli*, as previously described for other Cys-tagged Nbs [14,27,28].

### 4.4. Site-Specific Functionalization of Nb

Reduction: A solution of 2.5 mL (1.2 mg/mL, 37 µM, 3 mg) of dimerized Nb in PBS (pH 7.4) was reduced with 25 µL of a 0.5 M stock solution of ethylenediaminetetraacetic acid (EDTA) pH 7 and a 90-fold molar excess of 2-MEA (Acros Organics, Fisher Scientific, Merelbeke, Belgium). The reduction was completed after incubating the mixture at 37 °C for 90 min. The solution obtained after reduction was buffer-exchanged and purified on a PD-10 column pre-equilibrated with 0.2 M NH_4_OAc pH 6. Coupling of mal-NOTA(2,2′-(7-(2-((2-(2,5-dioxo-2,5-dihydro-1H-pyrrol-1-yl)ethyl)amino)-2-oxoethyl)-1,4,7-triazonane-1,4-diyl)diacetic acid (CheMatech, Dijon, France)): To the purified reduced Nb (typically 2.7 mg recovered in 3 mL) a 5-fold molar excess of mal-NOTA and 30 µL of a 0.5 M stock solution EDTA pH 7 were added. The mixture was incubated at 37 °C for 2 h, concentrated and purified by SEC.

QC of the end product was performed through SEC, SDS-PAGE and ESI-Q-ToF-MS. SPR was performed to assess the affinity, and the thermostability of the Nb was determined following procedures described in the Appendix A.

### 4.5. ^68^Ga-Labeling of the NOTA-mal-(hPD-L1) Nb

The NOTA-mal-(hPD-L1) Nb (3.6–4.6 µM) was diluted in 1 mL of 1 M NaOAc buffer pH 4.5–4.7, to which 1 mL of ^68^Ga-eluate (580–800 MBq) was added, and the mixture was incubated at 50 °C for 15 min. The radiolabeled Nb was purified on a PD-10 column pre-equilibrated with fresh injection buffer (0.9% NaCl, 5 mg/mL Vit. C, pH 6). The radiolabeled Nb was eluted in three different fractions of 0.5 mL, 2.5 mL and 0.5 mL. The highest activity fraction (2.5 mL) was filtered through a 0.22 µm (Millipore, Merck, Overijse, Belgium) filter. The RCP was measured before and after purification via iTLC. Decay-corrected radiochemical yield (DC-RCY) was calculated based on the activity recovered after PD-10 purification.

### 4.6. Stability Studies

The stability of the ^68^Ga-labeled Nb (15–50 MBq, after filtration) was tested over 4 h at RT and in human serum at 37 °C. RCP was assayed by iTLC and SEC. The samples were further diluted in 0.1 M sodium citrate buffer, 0.1% tween. The samples containing serum were filtered through a 0.22 µm filter before analysis.

### 4.7. Animal Models and Cell Lines

Dr. S.L. Topalian (National Cancer Institute, Baltimore, MD 21231, USA) provided the melanoma cell line HLA-A*0201^+^ 624-MEL. The 624-MEL cells were stably transduced to express hPD-L1, as previously described [12], and cultured in RPMI1640 medium supplemented with 10% Fetal clone I serum (Thermo Scientific, Merelbeke, Belgium), 2 mM L-Glutamine, 100 U/mL penicillin, 100 µg/mL streptomycin, 1 mM sodium pyruvate and nonessential amino acids. Female, five- to six-week-old athymic nude Crl:NU(NCr)-Foxn1nu mice were purchased from Charles River (France, Saint-Germain-sur-l’Arbresle). All experiments were performed in accordance with the European guidelines for animal experimentation under the license LA1230272. Experiments were approved by the Ethical Committee for the use of laboratory animals of the Vrije Universiteit Brussel (17-272-6). Intravenous injections were performed in the tail vein. Animals were anesthetized with 2.5% isoflurane in oxygen (Abbott Laboratories, North Chicago, IL 60064, USA) for injections, samplings, imaging and euthanasia.

### 4.8. Cell Binding Study

The radiolabeled Nb binding capacity was tested on hPD-L1^POS^ 624-MEL cells. Then, 5 × 10^4^ cells in 1 mL of medium per well were allowed to attach in a 24-well plate at 37 °C two days prior to the experiment. The plate was cooled to 4 °C 1 h prior to the experiment. The supernatant was removed, and the cells were incubated for 1 h at 4 °C with 500 μL of a 3 nM or a 15 nM radiolabeled Nb solution in unsupplemented medium (*n* = 3 wells per conditions). Unbound fractions were collected, and wells were washed two times with ice-cold PBS. Lysis of the cells was performed twice with 0.75 mL of 1 M NaOH at RT for 5 min. All fractions were collected and transferred to counting tubes to be measured in the γ-counter (Cobra Inspector 5003, Canberra-Packard, Schwadorf, Austria). Specificity was assayed on hPD-L1^NEG^ 624-MEL cells, and on hPD-L1^POS^ cells in the presence of a 100-molar excess of unlabeled competitor (unmodified Nb) following the same procedures. The percentage of bound activity was calculated as the measured cell-associated activity in the bound fractions divided by the activity of the added solution × 100.

### 4.9. Biodistribution and Tumor Targeting Studies

Female athymic nude mice (*n* = 6/group, experiment repeated for the hPD-L1^POS^ group) were injected subcutaneously in the right leg with 4.2 × 10^6^ hPD-L1^POS^ 624-MEL or hPD-L1^NEG^ 624-MEL cells. Tumor volume was measured twice weekly using an electronic calliper. The tumor volume was calculated using the following formula: (length × width^2^)/2. In about 30 days, tumors were allowed to reach a size of 296 ± 224 mm^3^ for hPD-L1^NEG^ tumors and a size of 129 ± 79 mm^3^ for hPD-L1^POS^ tumors. Xenografted mice bearing hPD-L1^POS^ or hPD-L1^NEG^ tumors were injected intravenously with 6 µg of [^68^Ga]Ga-NOTA-mal-Nb; 15.2 ± 1.5 MBq, 39.6 GBq/μmol or 15.4 ± 0.6 MBq, 39.9 GBq/μmol, respectively. Apparent molar specific activities are reported for the time of injection. The biodistribution was evaluated at 80 min p.i. After euthanasia, main organs and tissues were isolated, weighed and counted against a standard of known activity using a γ-counter. The amount of radioactivity in organs and tissues was expressed as a percentage of the injected activity per gram (% IA/g), corrected for decay. A single cell suspension from the tumors was prepared and FC analysis was performed to characterize hPD-L1 expression (procedure in the Appendix A).

### 4.10. Statistical Analyses

Results are expressed as mean ± SD. A non-parametric Mann–Whitney U test was carried out to compare data sets. Sample sizes and number of repetitions of experiments are indicated in the figure legends or in the materials and methods section. The number of asterisks in the figures indicates the statistical significance as follows: *, *p* < 0.05; **, *p* < 0.01; ***, *p* < 0.001; ****, *p* < 0.0001; Non-significant (NS).

## 5. Conclusions

In this study, we have confirmed that our lead Nb targeting human PD-L1 could be efficiently coupled to mal-NOTA via the cysteine–maleimide strategy, yielding a homogenous product for site-specific incorporation of the ^68^Ga-radionuclide. The ^68^Ga-labeling of NOTA-mal-conjugated Nb was efficient at 50 °C. The ^68^Ga-labeled Nb was stable and specific in vitro, and could specifically target hPD-L1 preclinically in vivo. Taken together, the ^68^Ga-labeled Nb via the mal–Cys chemistry is a promising PET imaging agent for future clinical assessment of PD-L1 expression.

## 6. Patents

Bridoux, J.; Broos, K.; Breckpot, K.; Xavier, C.; Lecocq, Q.; Keyaerts, M.; Devoogdt, N.; Raes, G.; Van Ginderachter, J. Human PD-L1-BINDING Immunoglobulins. No. PCT/EP2019/055133; WO/2019/166622; 6 September 2019.

## Figures and Tables

**Figure 1 pharmaceuticals-14-00550-f001:**
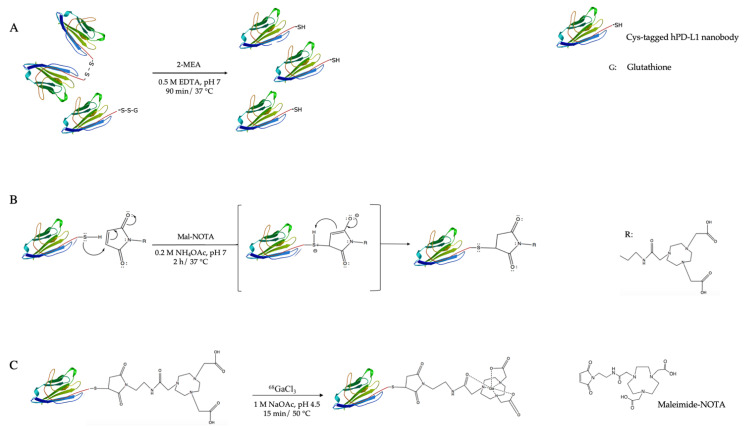
Representation of the functionalization and radiolabeling of the hPD-L1 Nb: (**A**) Dimerized Nbs are reduced in mild conditions (2-MEA = 2-mercaptoethylamine); (**B**) Site-specific functionalization of the Nb’s C-terminal cystein-tag with maleimide-NOTA via a Michael addition; (**C**) Gallium-68 labeling of the NOTA-mal-(hPD-L1) Nb.

**Figure 2 pharmaceuticals-14-00550-f002:**
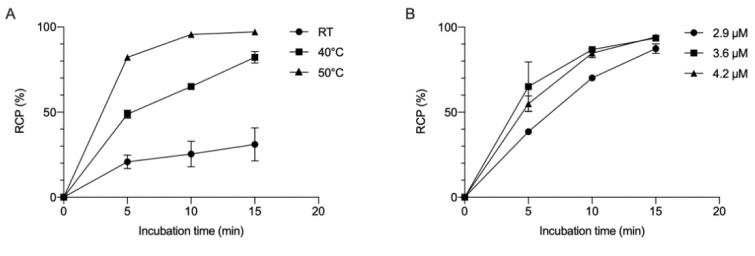
Optimization of the ^68^Ga-labeling of NOTA-mal-(hPD-L1) Nb: (**A**) Radiochemical purity (RCP) in %, expressed in function of the incubation time (min) at three different temperatures (room temperature (RT), 40 °C and 50 °C) at a constant Nb concentration of 3.6 μM. (**B**) Radiochemical purity (RCP) in %, expressed in function of the incubation time (min) at three different Nb concentrations (2.9 μM, 3.6 μM and 4.2 μM) at 50 °C.

**Figure 3 pharmaceuticals-14-00550-f003:**
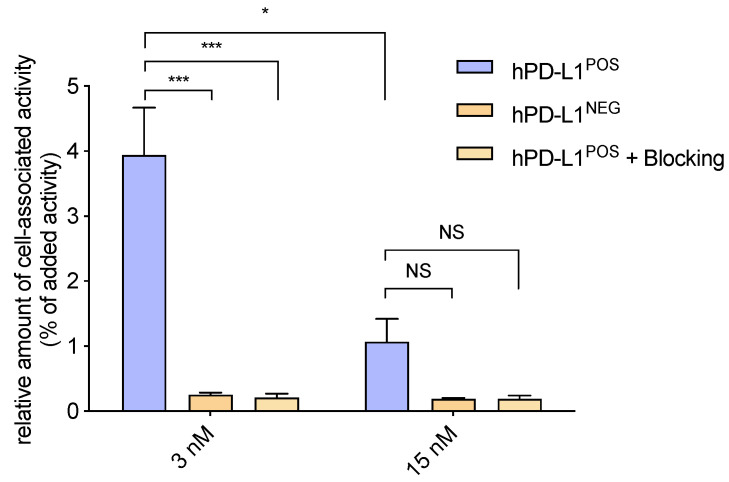
Relative amount of cell-associated activity of [^68^Ga]Ga-NOTA-mal-(hPD-L1) Nbs on hPD-L1^POS^ cells at 3 nM and 15 nM Nb concentrations, or on hPD-L1^NEG^ cells or in the presence of an excess of unlabeled Nb as control groups. (***, *p* < 0.001, *, *p* < 0.05, NS, non-significant).

**Figure 4 pharmaceuticals-14-00550-f004:**
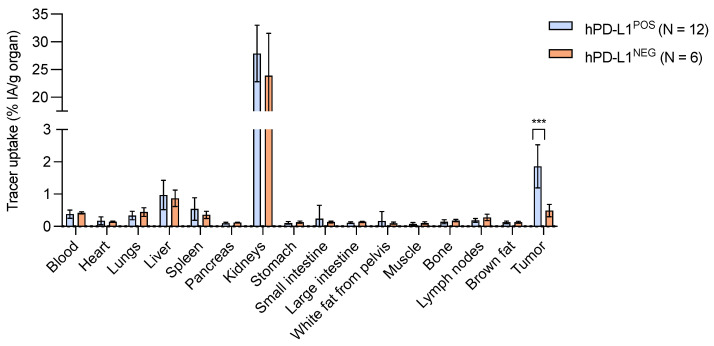
Biodistribution profile and tumor targeting of the [^68^Ga]Ga-NOTA-mal-(hPD-L1) Nb at 80 min post-injection in hPD-L1^POS^ tumor-bearing mice (*n* = 12) and in hPD-L1^NEG^ tumor-bearing mice (*n* = 6), showing a significant difference (***, *p* < 0.001) between hPD-L1^POS^ and hPD-L1^NEG^ tumor uptake.

## Data Availability

The data presented in this study are available on request from the corresponding author.

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
