# Peer review of "Site-Specific Radiolabeling of a Human PD-L1 Nanobody via Maleimide–Cysteine Chemistry"

_pharmaceuticals, 2021, doi:10.3390/ph14060550_

Round 1

Reviewer 1 Report

The manuscript  "Site-specific radiolabellig of a human PD-L1 nanobody via maleimide-cysteine chemistry" by Chigoho et al. present a very useful method for the site specific gallium-68 labelling of an interesting PET ligand. Binding studies and biodistribution show that the labelled Nb is a potentially useful radioligand in clinical research. The authors point out some weaknesses such as higher kidney uptake and lower production yield as compared to earlier presented Nbs. They also discuss  potential improvements how to overcome the mentioned limitations. 

This manuscript can be published in present form.

The experiments are well performed and described. 

Author Response

Dear reviewer,

We appreciate the time you took to read and review our article. We would like to thank you for your positive and motivating comments.

Reviewer 2 Report

This paper describes a valuable alternative to label Nanobodies with 68Ga site-specifically. The paper should be published after minor revisions.

Abstract, first sentence: “Immune checkpoint inhibitors targeting the programmed cell death-1 (PD-1) and its 14 ligand, PD-L1, have proven to be efficient cancer therapies in a subset of patients.” Both the ligand and the receptor are targets? Please check if that is correct.

Abstract: ….tend to respond better” Please add to what patients tend to respond better and clarify what you mean by “such information is not always used in treatment decision” Do you mean the knowledge that a patient’s tumor expresses PD-1?

Why is Nanobody written with a capital N?

Page 2: The authors suggest that PET could overcome the inability to test for the inhomogeneous distribution of PD-1/PD-L1 in the tumor environment. Please discuss the limits in resolution using 68Ga which arguably possesses a high positron energy. What resolution do the authors expect in tumors (especially small tumors)? Usually, the tumor’s microenvironment is hardly accessible via PET imaging. Please elaborate.

Page 2, line 68: omit “of” from “of yielding”

Page 2, line 95: change “controls” to “control”

Figure 1: when the authors talk about RCP do they actually mean radiochemical yield? If not please clarify.

Discussion: Please clarify if the [68Ga]Ga-NOTA-mal-(hPD-L1) Nb binds to the receptor or the native ligand. This is not clear from the provided information.

It would be very helpful to the reader if the authors would introduce a figure comparing the enzymatic and their new chemical approach to modify their NB. The additional visual demonstration would add clarity to the manuscript. This figure should show the different and important chemical groups involved in the NB derivatization.

Why did the authors not perform animal PET imaging?

Author Response

Dear reviewer,

Thank you for the relevant comments. We appreciate that you took the time to read and review our article. Here below you will find explanations and answers that will hopefully answer and clarify your questions and comments.

Reviewer 3 Report

The manuscript “Site-specific radiolabeling of a human PD-L1 nanobody via 2 maleimide-cysteine chemistry” aims to label an NB with 68Ga to target human PD-L1 using NOTA as chelating agent, conjugated via the cysteine-maleimide strategy. The authors obtained the 68Ga-labeled Nb with high yield and stability and showed its specificity for PD-L1 in vitro and in vivo.

This is an interesting, well-written and organized manuscript. However, there are some minor concerns:

Indicate the molecular weight of the unconjugated Nb in section 2.1.

The radiochemical purity of the labelled complex was analyzed only by ITLC and it should be better to use the higher resolution technique, HPLC. Include in the manuscript the Radio-HPLC of the product before and after purification or explain why this is not possible.

Explain why cells were lysed if the percentage of bound labelled complex can be determined in the whole cells.

In section 4.5, please indicate the volume of the PD-10 column.

Author Response

Dear reviewer,

Thank you for the relevant comments. I appreciate that you took the time to read and review our article. Here below you will find explanations and answers that will hopefully answer and clarify your questions and comments.
